# Effects of Doxorubicin Delivery by Nitrogen-Doped Graphene Quantum Dots on Cancer Cell Growth: Experimental Study and Mathematical Modeling

**DOI:** 10.3390/nano11010140

**Published:** 2021-01-08

**Authors:** Madison Frieler, Christine Pho, Bong Han Lee, Hana Dobrovolny, Giridhar R. Akkaraju, Anton V. Naumov

**Affiliations:** 1Department of Biology, Texas Christian University, Fort Worth, TX 76129, USA; madison.frieler@tcu.edu (M.F.); g.akkaraju@tcu.edu (G.R.A.); 2Department of Physics and Astronomy, Texas Christian University, Fort Worth, TX 76129, USA; christine.pho@tcu.edu (C.P.); bong.lee@tcu.edu (B.H.L.); h.dobrovolny@tcu.edu (H.D.)

**Keywords:** cancer, nanoparticles, *IC_50_*, graphene quantum dots, drug delivery, fluorescence, imaging, doxorubicin, mathematical modeling

## Abstract

With 18 million new cases diagnosed each year worldwide, cancer strongly impacts both science and society. Current models of cancer cell growth and therapeutic efficacy in vitro are time-dependent and often do not consider the *E_max_* value (the maximum reduction in the growth rate), leading to inconsistencies in the obtained *IC_50_* (concentration of the drug at half maximum effect). In this work, we introduce a new dual experimental/modeling approach to model HeLa and MCF-7 cancer cell growth and assess the efficacy of doxorubicin chemotherapeutics, whether alone or delivered by novel nitrogen-doped graphene quantum dots (N-GQDs). These biocompatible/biodegradable nanoparticles were used for the first time in this work for the delivery and fluorescence tracking of doxorubicin, ultimately decreasing its *IC_50_* by over 1.5 and allowing for the use of up to 10 times lower doses of the drug to achieve the same therapeutic effect. Based on the experimental in vitro studies with nanomaterial-delivered chemotherapy, we also developed a method of cancer cell growth modeling that (1) includes an *E_max_* value, which is often not characterized, and (2), most importantly, is measurement time-independent. This will allow for the more consistent assessment of the efficiency of anti-cancer drugs and nanomaterial-delivered formulations, as well as efficacy improvements of nanomaterial delivery.

## 1. Introduction

The incidence and mortality of cancer across the world is growing. In 2018, there were almost 18 million new cases of cancer diagnosed, along with 9.6 million deaths [1]. Among those, breast cancer is the most common cancer in women, and rates of breast cancer are expected to continue to rise in the future [1] despite the continuing development of cancer therapeutic strategies. Cancer is a disease of pathological hyperplasia of cells due to mutations that allow for traits such as self-sufficient growth signals, the evasion of apoptosis, and sustained angiogenesis [2]. Most common cancer treatment approaches include surgery, radiation, and chemotherapy. Many chemotherapies work by damaging DNA in cancer cells. For example, platinum-based therapeutics utilize alkylating agents to form covalent linkages in macromolecules [3], while doxorubicin damages cell DNA by disrupting topoisomerase-II (TOP2)-mediated DNA repair and generating free radicals to damage the cell membrane [4]. The replication of DNA and the transcription of RNA results in supercoiling of DNA strands. TOP2a and TOP2b create paired DNA strand breaks. The critical intermediate for this reaction is the TOP2 cleavage complex. Topoisomerase II inhibitors such as doxorubicin stabilize this complex, which leads to an accumulation of double-stranded breaks as relegation is inhibited. This results in the accumulation of DNA damage, leading to cell death in rapidly dividing cells [5].

One major issue affecting a number of current chemotherapies, including DNA-damaging drugs such as doxorubicin, is their lack of ability to differentiate between normal and cancer cells. These therapeutics target any dividing cell. While doxorubicin may affect cancer cells at a higher rate, healthy cells will still perish, thus leading to a number of side-effects including cardiovascular toxicity, which can lead to hypotension, tachycardia, arrythmias, and congestive heart failure [6]. The incidence of doxorubicin cardiomyopathy can be as high as 36% when a person’s lifetime dose exceeds 600 mg/m^2^. Doxorubicin accumulation in the mitochondria of cardiac cells may lead to increased oxidative stress in the cell and also contribute to the mitochondrial permeability transition pore opening, thus inhibiting the ability of the inner membrane of the mitochondria to maintain an electrochemical gradient [6]. These substantial drawbacks of the widely used therapeutic approach can be mitigated by focusing its delivery to tumors and/or decreasing its toxicity for healthy tissues. Nanoparticle delivery can alleviate the dose-dependent toxicity of chemotherapy drugs by decreasing the treatment dose by enhancing accumulation and/or efficacy [7]. Additionally, nanoparticles provide a platform for targeting agents that, by binding preferentially to particular receptors on cancer cells, can focus the delivery to tumor sites. The intrinsic fluorescence of some nanoparticles also offers the possibility to track chemotherapy delivery in cells and tissues. The photostable fluorescence emission arising in a number of nanomaterial classes due to quantum confinement and edge effects may allow one to monitor the activity of the drug and its release into the tissues in real time [8]. A variety of different nanoparticles have shown promising potential in anticancer drug delivery and imaging, including inorganic quantum dots [9], silicon nanomaterials [10], silver nanoparticles [11], liposomes [12], and gold nanoparticles [13]. Gold nanoparticles include nanoshells [14], nanorods [15], and other nanoscale gold constructs utilized for drug delivery [16], early cancer cell identification by EGFR (epidermal growth factor receptor) overexpression [17], or Raman-based detection of epithelial cancers [18].

Another, perhaps somewhat underexplored for bioapplications, class of nanomaterials includes nanocarbon platforms. Recently, some carbon allotropes were utilized for drug delivery because of the unique properties they possess. For instance, quasi-one-dimensional carbon nanotubes allow for the attachment of a high density of drugs because of their high surface area-to-volume ratio, ability to non-covalently complex nucleic acids for gene delivery, [19,20,21], and apparent ability to evade the immune response [22]. They also exhibit near-infrared fluorescence, thus allowing for high penetration depth in vivo fluorescence imaging in animal models [23]. Zero-dimensional graphene or carbon quantum dots may be selectively synthesized to render higher biocompatibility [24] and solubility in water [25], or they may be functionalized to provide prolonged circulation time in the body [26] or degrade in biological media [27]. Additionally, a variety of graphene and carbon quantum dots exhibit confinement-originating intrinsic fluorescence that can be utilized to track those within the biological cells and tissues and, therefore, follow drug administration [28,29,30]. These properties allow such carbon nanomaterials to address a variety of applications in medicine from imaging and sensing to treatment. In our recent studies [27], graphene quantum dots developed in a bottom–up approach from glucosamine showed a high cellular uptake even without targeting, while similar graphene quantum dot structures were actively taken up by precision cut mammary tumor slices with no obvious negative effects to the tissue [8]. Graphene quantum dots have been shown to focus drug delivery on the cell nucleus [31] and enhance the DNA cleavage activity of doxorubicin [31,32]. Thus, the use of these nanoparticles in conjunction with known chemotherapies may lead to the reduction of the harmful high dose side effects of chemotherapy drugs [31], while their intrinsic fluorescence allows for the real-time monitoring of drug accumulation and can help monitor delivery pathways [33]. There are ongoing attempts to enhance therapeutic efficacy with nanomaterial delivery, and facilitate fluorescence image tracking with a single biocompatible material. In this work, we utilized novel, highly biocompatible nitrogen-doped graphene quantum dots (N-GQDs) [24] that underwent rigorous characterization as a potential delivery/imaging/sensing agent [27]. They exhibited high quantum yield (>62%) visible and near-infrared fluorescence, biocompatibility at over 1 mg/mL concentrations, and apparent biodegradability in a cell culture. The combination of these remarkable properties make them an ideal candidate drug delivery vehicle. Here, N-GQDs served as a delivery platform for a conventional therapeutic, doxorubicin (DOX), while we assessed the improvement in therapeutic efficacy using a novel mathematical model we developed to describe cell growth.

Mathematical models are increasingly being used to personalize [34,35,36] treatment regimens for patients. While this practice will eventually require complex mathematical models that include many biological processes [37,38], simpler models can still help provide insight into cancer treatment. The mathematical modeling of drug effect in cancer cells allows for the prediction of the doses required for both in vitro and further in vivo treatment, as well as the assessment of their expected efficacy [39,40]. Simple models have already been used to make predictions about the effectiveness of cancer treatments [41,42], including combination therapies [39,43]. Current strategies for modeling efficacies of drugs alone use the *IC_50_* (concentration of the drug at half maximum effect) measurement and, less commonly, the *E_max_* (the maximum reduction in the growth rate [11]) measurement to give the correct drug dosage to a patient. The real time-dependent nature of these curves [44] can lead to inaccuracies in theoretical models due to their bias toward exponential growth and delays in drug effect stabilization, [44,45], thus requiring new modeling approaches that can estimate time-independent parameters.

Thus far, mathematical modeling approaches have been developed for the drug delivery of nanomaterials based on hydrogel [46], mesoporous silica [40,47], and pegylated gelatin [48] that model for time-dependent drug concentration and tumor volume equations followed by the nanotherapy [49]. Such approaches are novel because they optimize the properties of the nanomaterial that have been often limited by clinical data and nanotherapeutics for clinical use [50], allowing for one to increase efficacy and/or repurpose existing therapeutics [51]. The present study, however, presents a novel method of modeling drug delivery via a time-independent approach and, for the first time, provides a model for cancer therapeutic delivery by graphene quantum dots. While GQDs have previously been experimentally conjugated with DOX for cancer-targeted drug delivery [32,52], the N-GQDs utilized in this work possess higher biocompatibility and allow for multicolor visible/near-infrared fluorescence imaging [27]. As a result, this joint theoretical/experimental work aimed to create a new time-independent mathematical modeling technique to provide more consistent *IC_50_* values to use in the dosage of drugs based on experimental cell counting and cytotoxicity assay data as we explored the delivery of doxorubicin with N-GQDs. We tested our model to describe the effects of this nanomedicine on cancer cell growth and showed a substantial decrease in *IC_50_* of the drug that could ultimately lead to decreased chemotherapy side effects in patients through the utilization of lower levels of chemotherapeutics with nanomaterial delivery.

## 2. Materials and Methods

### 2.1. Cell Culture

MCF-7 (breast cancer cells), HEK293 (human embryonic kidney cells), and HeLa (cervical cancer cells) cells were grown in cell culture using Dulbecco’s Modified Eagle Media (DMEM; 10% fetal bovine serum, 1% non-essential amino acids, L-glutamine, penicillin (100 U/mL), and streptomycin (100 mg/mL)) at 37 °C and 5% CO_2_. When cells reached confluency, the medium was aspirated, cells were washed with 1× PBS (phosphate buffered saline). 0.05% trypsin was added to detach cells, and the culture was quenched with DMEM. Cells were counted using a hemocytometer, plated, and then used in an experiment; meanwhile, 10% of cells were added to a new flask with DMEM.

### 2.2. Cell Growth Assay

MCF-7, HEK293, and HeLa cells were plated on 12-well trays at a density of 500, 1000, or 2000 cells/well with 2 mL of media in each well, depending on the experiment. The plates were incubated at 37 °C and 5% CO_2_. After 24 h, the drug was added at concentrations 0.05, 0.005, 0.0005, and 0.00005 μg/mL. After another 24 h, the medium was removed from 3 of the wells and washed with 1× PBS. Then, 0.05% trypsin was added to detach the cells, and the medium was added to quench the trypsin. Cells were then counted using a hemocytometer, and the steps of removing the medium, detaching cells, and counting were repeated for each concentration of the drug.

### 2.3. MTT Cell Viability Assay

Cells were plated at a density of 5000 cells/well on a 96-well plate and incubated at 37 °C and 5% CO_2_. After 24 h, the cells are treated with increasing concentrations ranging from 0.00625 to 1.6 μg/mL of doxorubicin or with DOX-N-GQDs with an equivalent DOX content. The cells were treated with the drug for 16 h, and then the medium was removed from the wells. A solution of 1 mg/mL MTT (thiazolyl blue tetrazolium bromide) in a serum-free medium was made, after which 100 µL were added to each well and incubated at 37 °C for 4 h. Then, the MTT solution was removed and 100 µL of DMSO (dimethyl sulfoxide) were added. Cells were then placed on a shaker for 5 min, and the absorbance was measured at 540 nm in a FLUOstar Omega microplate reader (BMG Labtech, Cary, NC, USA) and analyzed with Omega software (BMG Labtech, Cary, NC, USA).

### 2.4. Synthesis of DOX-N-GQDs

A solution of DOX-N-GQDs was made by combining 1 mL of DOX at a concentration of 0.0041 mg/mL with an aqueous suspension of N-GQDs at a concentration 1 mg/mL that was synthesized from a single glucosamine-HCl precursor via a hydrothermal microwave treatment, as described in [53]. This suspension was ultrasonically treated in a ultrasonic disperser (Covaris, Woburn, MA, USA) for 7 cycles of 30 s bursts at a power of 1 W to ensure the non-covalent complexation of the drug.

### 2.5. Microscopy

A semi-motorized inverted microscope (IX73P2F, Olympus, Center Valley, PA, USA) was used for fluorescence confocal microscopy. The sample slides were excited by a mercury lamp (U-HGLGPS, Olympus, Center Valley, PA, USA) with a 480 ± 20 nm excitation filter. For detection, a CMOS (complementary metal oxide semiconductor, Prime 95B, Photometrics, Tucson, AZ, USA) camera was utilized with a 600 ± 27.5 nm emission filter at an integration time of 200 ms. Voxel images were generated using the disk scan unit (DSU) confocal add-on (IX2-DSU/BX-DSU, Olympus, Center Valley, PA, USA): with an automated z-stacking imaging capability, 20–30 z-slice images were acquired for the voxel image using a step size of 0.35 μm, which was the limiting threshold. A 460 ± 20 nm excitation filter was used for these voxel images.

### 2.6. Image Analysis

The ImageJ software (Bethesda, WA, USA) was used to quantify the fluorescence in the microscopy images. Cells were outlined with a freehand selection in the software, and the integrated density was measured. To obtain the corrected total cell fluorescence (CTCF), the background of the images was measured to assess the mean fluorescence background. CTCF was calculated as the difference of the integrated density and the product of the area of the outlined cell and the mean fluorescence background.

### 2.7. Mathematical Modeling

We used a logistic growth model to describe growth of cells in the absence of any drug treatment:$$\frac{dN}{dt}=\lambda N\left(1-\frac{N}{K}\right),$$
where *N* is the number of cells, *λ* is the cell growth rate, and *K* is the carrying capacity of the well. We assumed that when doxorubicin was applied, either on its own or delivered via N-GQD, it decreased the growth rate of the cells. This was modeled through the use of drug efficacy, *ε*, which was related to drug concentration via the *E_max_* model [54]:$$\varepsilon =\frac{E_{max}D}{D+IC_{50}},$$
where *D* is the applied drug concentration, *IC_50_* is the drug concentration that achieves half the maximum effect, and *E_max_* is the maximum reduction in growth rate. We multiplied the growth rate by (1 − *ε*) to reproduce growth curves at different drug concentrations.

We fit the model to the data in a two-step process, first fitting the control data to get estimates for *λ* and *K*. We then simultaneously fit the remaining growth curves with treatment to estimate *IC_50_* and *E_max_*. Fitting was performed by minimizing the sum of squared residuals (SSR) using the Nelder–Mead algorithm implemented through the minimize function in Python’s SciPy package. Confidence intervals were determined by performing 1000 bootstrap replicates.

## 3. Results

### 3.1. Cell Growth with DOX Treatment

In order to assess the direct effect of chemotherapy drugs on the cell growth rate, we set the baseline and monitored cell growth rate of three different cell types: HeLa cells, a cervical cancer cell line; MCF-7, a breast cancer cell line; and non-cancerous control HEK293 cells, human embryonic kidney cells obtained by transforming primary cells with adenovirus DNA. Each cell type was grown over a period of 14 days, with cells counted every other day (Figure 1a). The control for the experiments in this work provided by the growth analysis of the untreated cells showed HeLa and HEK293 cell growth peaking at day 10, while the MCF-7 cell growth peak was slightly delayed to day 12. The subsequent growth decline was likely due to the overcrowding of the wells and a buildup of waste in the medium. After establishing a control baseline for cell growth, the cytotoxic effect of doxorubicin was further assessed. HEK293, HeLa, and MCF-7 cells treated with doxorubicin at the concentrations 0.05, 0.005, 0.0005, and 0.00005 μg/mL showed varying cell growth levels (Figure 1b–d, respectively). The highest concentration of doxorubicin used (0.05 μg/mL) proved to be too toxic to the cells, resulting in no measurable cell growth in all cell lines tested. In HEK293 cells, all concentrations other than 0.05 μg/mL showed a similar rate of growth, including the cells that had no drug added. HeLa cells showed varying growth rates per concentration, while MCF-7 cells exhibited a decreased growth rate as the concentration of doxorubicin increased. In order to supplement cell growth rate data with bulk cell viability, information of the percent of surviving cells was evaluated with the MTT cell viability assay. This study showed a concentration-dependent decline in cell viability with a calculated coefficient of determination (R^2^) value of 0.712 for HEK293 cells, 0.955 for MCF-7 cells, and 0.983 for HeLa cells (Figure 1e), thus indicating the proportion of variance in cell death due to doxorubicin toxicity. In the cell growth experiments, the cells were treated and counted for 14 days, while in the cell viability assay, the cells were only treated for 16 h. Therefore, the longer exposure during the cell counting experiments necessitated a lower concentration range than the MTT cell viability assay to avoid total cell death. It was anticipated that DOX treatment would have different effects in different cancer cell lines, in part due to the differences in their replication rates. However, both cancer cell lines exhibited a DOX concentration-dependent cell viability decrease, with a less steep trend seen in the HEK293 cells.

### 3.2. Characterization of DOX-N-GQDs

In this work, DOX was non-covalently bonded to N-GQDs through the π-stacking of an aromatic DOX platform onto N-GQD’s graphitic backbone. GQDs were characterized by the TEM as 3–5 nm quasi-spherical structures with discernable graphitic lattice fringes (Appendix A). While EDX (energy dispersive X-ray spectroscopy, JEOL, Peabody, MA, USA) showed the presence of carbon, oxygen, and nitrogen within the GQDs (Appendix A), their FTIR spectra possessed a number of features characteristic of the oxygen and nitrogen-containing functional groups decorating the surface of the GQDs. (Appendix A). The DOX-N-GQD complexes were also characterized by the absorption and fluorescence spectroscopies to assess complexation, concentration, and the capability of fluorescence tracking. As seen in Figure 2a, the absorption of the complex showed peaks that indicated the presence of both doxorubicin and graphene quantum dots. The analysis of the ~500 nm peak present only due to DOX allowed for the assessment of DOX loading. All concentrations of N-GQD-DOX are therefore described in this work by their content of DOX for comparison with the drug alone. Additionally, the fluorescence of the complex, as seen in Figure 2b, only showed the emission features of the doxorubicin, while N-GQDs were quenched via charge transfer from the attached DOX molecules. DOX fluorescence was, therefore, further utilized to track the drug delivery in vitro.

DOX-N-GQD complexes were visualized in the cells using confocal fluorescence imaging at 3, 6, 9, 12, and 24 h time points after introduction into the cell media (Figure 3a–e) observed in confocal 3D cell images. Already at 3 h, substantial DOX emission was observed from cell nuclei. Lower intensity N-GQD fluorescence was also detected in the cells, thus suggesting their successful internalization as DOX.

It is critical to point out that DOX fluorescence was mainly originating from cell nuclei with minimal emission from the cytoplasm, whereas low intensity GQD green emission (Appendix A) was observed throughout the cells. The efficiency of DOX-N-GQD internalization was further assessed via the intensity of DOX fluorescence per unit cell area. This was described by the CTCF for each time point (3, 6, 9, 12, and 24 h) averaged over 200 cells (Figure 3f). CTCF calculations indicated the highest DOX content in the cells at 9 h post treatment, while its level stayed high for up to 24 h.

### 3.3. Treatment of Cells with N-GQD-DOX Complex

The efficacy of the N-GQD-delivered DOX was further evaluated via two separate assays: cell growth analysis and an MTT cell viability assay. The MTT assay served to verify both the efficacy of the treatment and the biocompatibility of the N-GQDs, thus also verifying their possible contribution to the toxic effect of the combined DOX-N-GQD formulation. The concentration of N-GQDs was matched to that in DOX-N-GQD complexes with corresponding DOX loading to allow for comparative analysis. As a result, N-GQDs alone showed very little toxicity to cells even at the highest concentration of 1 mg/mL, equivalent to DOX loading of 1.6 μg/mL, while the DOX-N-GQD formulation was shown to be more effective in treatment than doxorubicin alone (Figure 4a). N-GQDs alone at lower concentrations could even enhance cell viability because they can be degraded and metabolized by the cells [27].

The direct effects of DOX-N-GQD conjugates on cancer cell growth were further evaluated via cell counting experiments (Figure 4b) and compared to the effects of DOX alone (Figure 1). At concentrations of 0.005 and 0.05 μg/mL, cells treated with DOX-N-GQDs did not exhibit growth, thus indicating that these concentrations were too toxic. At concentrations of 0.0005 and 0.00005 μg/mL of DOX-N-GQDs, cell growth was progressively inhibited with DOX dose increases. Even though some concentrations did not achieve 100% cell death, the collected data were sufficient for modeling.

The obtained experimental cell growth rate data were used to model the growth of the cells following treatment with chemotherapy drug with or without the N-GQD delivery (Figure 5a), as described by a logistic model:(1)$$\frac{dN}{dt}=\lambda N\left(1-\frac{N}{K}\right),$$

In this model, *N* is the number of cancer cells, *λ* is the growth rate, and *K* is the carrying capacity. We assumed that doxorubicin would reduce the cell growth rate in a time-independent manner. The growth of MCF-7 cells after treatment with DOX-N-GQDs was further modeled using the same equation (Figure 5b). A comparison of the *IC_50_* and *E_max_* in cells treated with doxorubicin alone versus those treated with varying concentrations of DOX-N-GQDs (Figure 5c) elucidated the decreased *IC_50_* and an increase in the *E_max_* value for DOX-N-GQD conjugate treatment. These changes were indicative of an increased drug efficacy with N-GQD delivery.

A measurement of cell growth based on cell count can be prone to some degree of observer error. To account for that, a different method (MTT assay) was used to measure cell death on days two, four, six, and eight to assess the growth rate of MCF-7 cells in the presence of doxorubicin and DOX-N-GQDs. These measurements showed a steady increase in cell death rate for both treatments over the eight-day period, as shown in Figure 6. The *IC_50_* for the DOX-N-GQDs treatment of 0.97 μg/mL (with respect to DOX concentration) appeared to be substantially below the 1.77 μg/mL *IC_50_* for DOX alone. These results resembled the results of the mathematical models of cell growth described above. We also point out that at low concentrations of the drug, cell growth over time could lead to increased cell numbers, even for longer treatments.

We further used the parameters determined from the aforementioned model fit to the cell growth data (Figure 5c) to simulate the MTT assays (Figure 7a). The simulations produced curves resembling those observed experimentally (Figure 7b), with increasing cell death rate over the eight-day period. The *IC_50_* values obtained from the simulated data (Figure 7c) decreased with increasing measurement time and indicated that DOX-N-GQDs were more efficacious than doxorubicin alone.

## 4. Discussion

In this work, we explored the delivery of DOX with N-GQDs, assessed the advantages of this formulation in vitro, and developed models to describe cell growth in the presence of the nanomaterial-delivered therapeutics. A baseline of cell growth established without treatment was used to assess the effects of doxorubicin in HeLa, MCF-7, and HEK-293 cells. It was evident (Figure 1) that no cell growth occurred at any DOX concentration above 0.05 μg/mL due to high toxicity of that amount of the drug to all cell lines. In HEK293 and HeLa cells, concentrations below 0.05 μg/mL of doxorubicin showed no statistically significant difference in cell growth; however, MCF-7 cells exhibited a decline in cell growth as the drug concentration increased, indicating a dose-dependent response to the drug (Figure 1d) expected from cancer cells. Complementary to cell counting, cell viability studies assessing DOX efficacy indicated that MCF-7 and HeLa cells were affected at a higher rate than the non-cancer HEK293 cells. Both HeLa and MCF-7 cells are rapidly dividing cancer cells and, therefore, could be more sensitive to DOX. Because of their distinct dose-dependent response to doxorubicin, we identified the MCF-7 cell line as a suitable platform for subsequent experiments with DOX-N-GQD complexes.

The absorption spectra of the non-covalently bound DOX-N-GQDs indicated the presence of both doxorubicin and N-GQDs in the compound. A ~500 nm peak presented a characteristic feature of DOX, while peaks near 250 and 300 nm were indicative of the presence of N-GQDs (Figure 2a). However, a fluorescence spectrum of the DOX-N-GQD complex indicated the substantial quenching of N-GQDs by doxorubicin. This could be attributed to strong electronic coupling and charge transfer interactions between DOX and the N-GQDs. Furthermore, coupling with N-GQDs was shown to substantially increase the emission intensity of DOX (Figure 2b), likely due to a higher N-GQD absorption cross-section. This verified the strong interaction between the nanomaterial and the drug.

The confocal imaging of the MCF-7 cells after treatment with DOX-N-GQDs showed the localization of doxorubicin, mainly in the cell nucleus, while the full colocalization study for the NGQD internalization involving the staining of the cell compartments was performed in our previous work [27]. Localization in the nucleus enabled the proposed mechanism of action of doxorubicin, because the target of this drug, a topoisomerase II inhibitor involved in DNA replication, was present in the cell nucleus. Furthermore, green N-GQD emission was observed post internalization, thus indicating a successful release of the quenching drug. A time course experiment measuring the efficiency of internalization of doxorubicin into the cells indicated the increase in DOX content up to 9 h that remained in the cells for at least 24 h following treatment, allowing the drug to efficiently induce cancer cell death. This effect was evaluated via cytotoxicity measurements showing that N-GQDs alone showed almost no toxicity in the cells, with a slight increase in cell viability attributed to N-GQD metabolization by the cells in the previous works [27]. However, the DOX-N-GQD conjugate appeared more toxic to the cells than doxorubicin alone with an over 1.8 factor difference in the *IC_50_* on day two. This constituted an improvement of the therapeutic efficacy by biocompatible nanomaterial delivery and outlined an advantage of using nanoscale platforms for drug transport. This was further confirmed via cell counting studies that helped model the direct drug effect on cell replication. Unlike for DOX alone (Figure 1), where only a 0.05 μg/mL concentration showed complete cell death, DOX-N-GQDs exhibited such an effect for both 0.005 and 0.05 μg/mL DOX contents, with dose-dependent therapeutic responses for 0.0005 and 0.00005 μg/mL (Figure 4). Since N-GQDs did not contribute to the toxic response at the concentrations used, N-GQD delivery facilitated a more substantial DOX therapeutic effect that allowed for the complete suppression of cancer cell growth at lower concentrations than for DOX alone. DOX-N-GQDs provided a lower *IC_50_* assessed from the MTT assay by an average of 1.76 times compared to DOX alone (Figure 7b). Additionally, it is critical that a concentration of 0.005 μg/mL of DOX when delivered by the N-GQDs resulted in the same inhibitory effect as 0.05 μg/mL of DOX alone (Figure 1d or Figure 4b). This indicated that a therapeutic could be used at 10 times lower doses when delivered by the N-GQD, which could help circumvent its adverse effects to the body.

The mathematical modeling of the cell growth data rendered simulated growth curves similar to those obtained by cell counting both for DOX alone and DOX-N-GQDs. The model was able to accurately reproduce the growth rate of MCF-7 cells after they were treated with doxorubicin and/or nanomaterial-derived therapeutic, and it allowed us to compute the time-independent values of *IC_50_* and *E_max_*. Complementary to the advantage of utilizing lower DOX doses with N-GQD delivery, as highlighted above, this model showed 5.4 times lower *IC_50_* and 1.1 times elevated *E_max_* values for DOX-N-GQDs compared to those of DOX alone (Figure 5c), thus predicting lower needed doses and higher efficacy for DOX-N-GQDs.

When an alternative MTT assay experimental validation (Figure 7b), utilizing cell death to assess drug efficacy, was introduced to account for the potential discrepancies of previously used cell counting assays, it also confirmed the decreased *IC_50_* for the DOX-N-GQDs treatment compared doxorubicin alone. Our mathematical model (Figure 7c) was further able to reproduce the trends seen in the MTT assays, showing a reduced *IC_50_* (Figure 7a) for the DOX-N-GQD conjugate with lowered *IC_50_* values at later measurement days. This predicted assay (Figure 7a) was qualitatively consistent with experimental findings and showed a similar trend to the one observed experimentally (Figure 7b). This verified both the model sustainability and the advantages of the N-GQD-based therapeutic delivery.

## 5. Conclusions

In this work, we performed cell counting and MTT assay studies to outline the advantages of the doxorubicin chemotherapeutic delivery by nitrogen-doped graphene quantum dots, and we utilized these data to develop a model to estimate time-independent *IC_50_* and *E_max_* values of the therapeutics. The lower 0.97 μg/mL *IC_50_* of DOX-N-GQDs, as compared to 1.77 μg/mL *IC_50_* for the DOX alone, indicated the improved efficacy of the GQD-delivered cancer therapeutic payload. Utilizing N-GQDs as a carrier also often allowed for the use of 10 times lower doses of DOX for the treatment of the same efficacy with no additional toxic effect from N-GQDs. This therapeutic combination could be traced via DOX fluorescence enhanced by the N-GQD platform. Furthermore, upon DOX release, the observed restoration of N-GQD green fluorescence previously quenched by interaction with DOX indicated successful therapeutic clearance.

Based on these experimental studies, we developed a novel mathematical model of cell growth in the presence of the therapeutic that could accurately portray the rate of cancer cell growth after treatment with DOX or DOX-N-GQDs. While the logistic model neglects many of the biological details of the cancer growth process [55], it has few parameters, thus allowing it to focus its parameter estimation on the effect of the treatment. Modeling suggested a 10% increase in the maximum efficacy of drug treatment when DOX was delivered by N-GQDs. Model parameter estimates also indicated that the *IC_50_* decreased by 82% when DOX was delivered by N-GQDs. The results of the model were confirmed by the independent cell viability study, again showing lower *IC_50_* values for DOX-N-GQD treatment.

This joint experimental/modeling approach not only presents an improved chemotherapeutic efficacy via the delivery by a biocompatible N-GQD platform but also gives a predictive power for describing nanomaterial-based drug delivery effects in a time-independent method that has not been developed to date. These developments will allow for more consistent measurements of the efficacy of chemotherapy nanoformulations and to highlight the possibility of using lower dosages with nanomaterial delivery to decrease the risk of side effects from high doses of chemotherapy.

## Figures and Tables

**Figure 1 nanomaterials-11-00140-f001:**
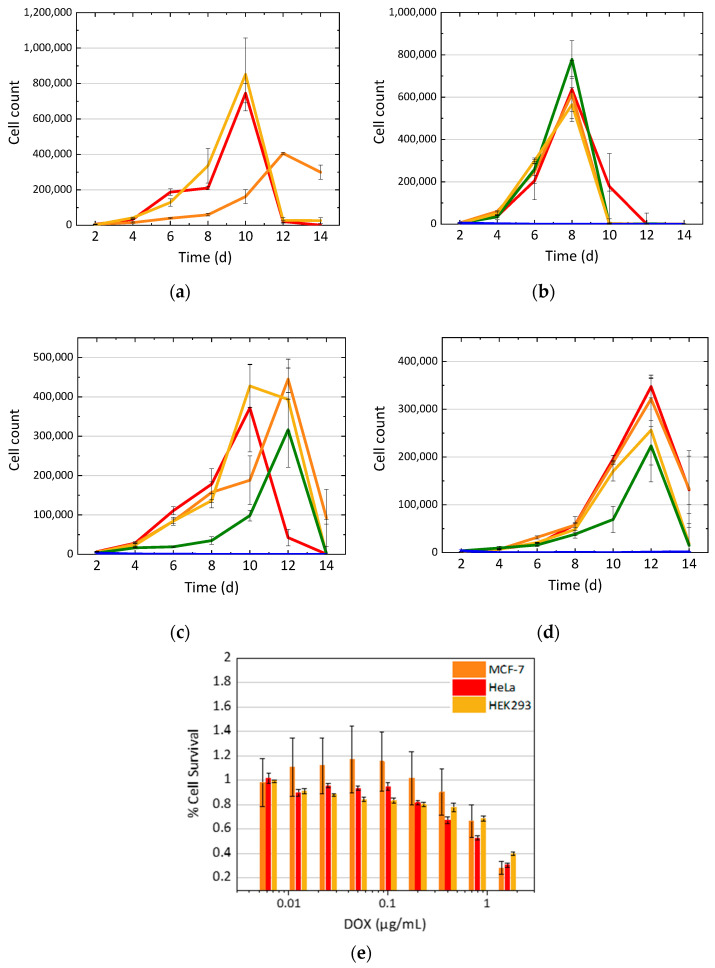
(**a**) Growth of HeLa (red), MCF-7 (orange), and HEK293 (yellow) cells over 14 days. (**b**) HEK 293, (**c**) HeLa, and (**d**) MCF-7 growth curves after treatment with doxorubicin at 0 mg/mL control (red), 0.00005 μg/mL (orange), 0.0005 μg/mL (yellow), 0.005 μg/mL (green), and 0.05 μg/mL (blue) concentrations. (**e**) Cytotoxicity of doxorubicin in MCF-7 (orange), HeLa (red), and HEK 293 (yellow) cells, evaluated via a thiazolyl blue tetrazolium bromide (MTT) cell viability assay.

**Figure 2 nanomaterials-11-00140-f002:**
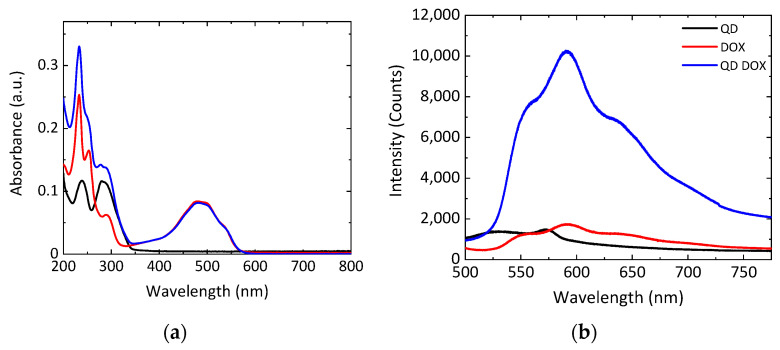
Nitrogen-doped graphene quantum dot (N-GQD) (black), doxorubicin (DOX) (red), and DOX-N-GQD (blue) complex absorption (**a**) and fluorescence (**b**) spectra.

**Figure 3 nanomaterials-11-00140-f003:**
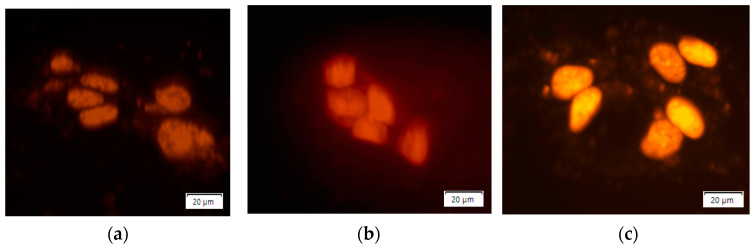
(**a**) Confocal 3D z-stack fluorescence images of DOX-N-GQD complex in MCF-7 cells at (**a**) 3 h, (**b**) 6 h, (**c**) 9 h, (**d**) 12 h, and (**e**) 24 h time points. Emission mainly originated from cell nuclei. (**f**) Variation of average fluorescence of doxorubicin per cell unit area in MCF-7 cells with time reflecting internalization dynamics.

**Figure 4 nanomaterials-11-00140-f004:**
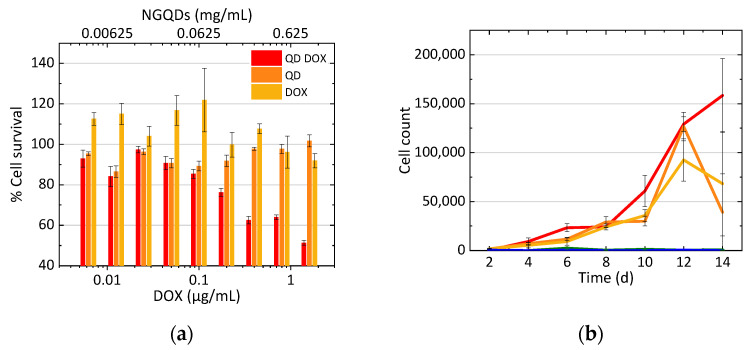
(**a**) Cytotoxicity of DOX (yellow), N-GQDs (orange), and DOX-N-GQDs (red) in MCF-7, as evaluated via an MTT cell viability assay. (**b**) MCF-7 cell growth curve up to 14 days after treatment with DOX-GQDs with DOX concentrations of 0 μg/mL (N-GQDs alone) (red), 0.00005 μg/mL (orange), 0.0005 μg/mL (yellow), 0.005 μg/mL (green), and 0.05 μg/mL (blue).

**Figure 5 nanomaterials-11-00140-f005:**
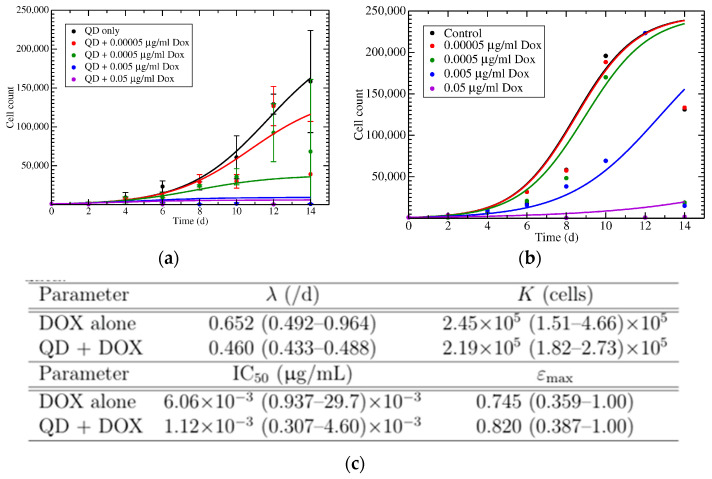
(**a**) Mathematical model fit to growth data of MCF-7 cells treated with DOX-N-GQDs. (**b**) Mathematical model fit to growth data of MCF-7 treated with DOX alone. (**c**) Best fit parameters with 95% confidence interval for MCF-7 cell growth.

**Figure 6 nanomaterials-11-00140-f006:**
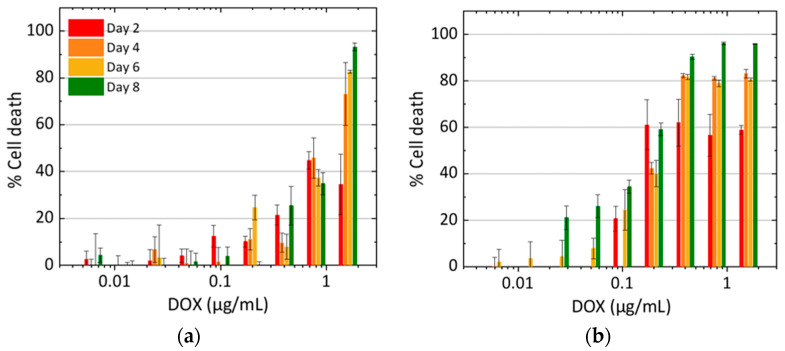
Rate of MCF-7 survival after (**a**) DOX, (**b**) DOX-N-GQD treatment at day 2 (red), day 4 (orange), day 6 (yellow), and day 8 (green).

**Figure 7 nanomaterials-11-00140-f007:**
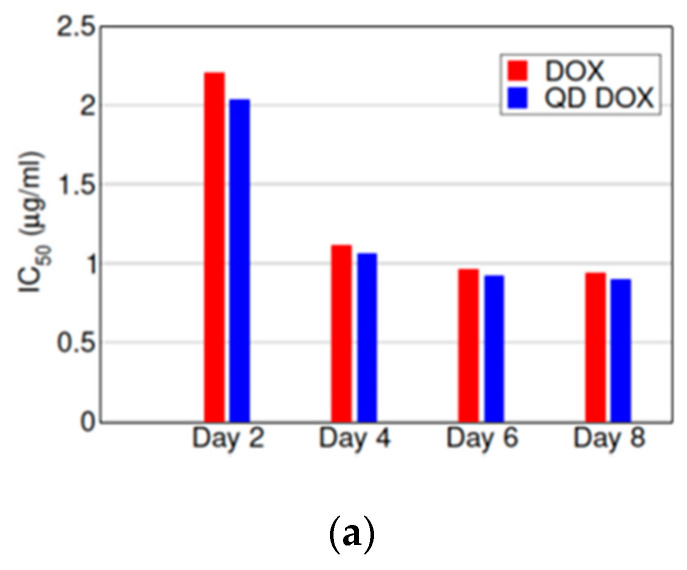
(**a**) *IC_50_* determined from the simulated MTT assays. (**b**) *IC_50_* (concentration of the drug at half maximum effect) for DOX (red) or DOX-N-GQD (blue) treatment in MCF-7 cells. (**c**) Simulation of the MTT assay for doxorubicin alone (circles) and DOX-N-GQDs (squares) using the fit parameters in Figure 5c.

## Data Availability

The data presented in this study are available on request from the corresponding author. The data are not publicly available due to privacy.

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
