# Peer review of "Effects of Doxorubicin Delivery by Nitrogen-Doped Graphene Quantum Dots on Cancer Cell Growth: Experimental Study and Mathematical Modeling"

_nanomaterials, 2021, doi:10.3390/nano11010140_

Round 1

Reviewer 1 Report

This is a very interesting study by Frieler et. al.. The authors assess the efficacy of Doxorubicin chemotherapeutic and compare the delivery method with and without nitrogen-doped graphene quantum dots (N-GQDs). The use of N-GQDs allowed using up to 10 times lower drug doses with the same therapeutic effect. The study is accompanied by a model of cancer cell growth. The work is well presented and the paper is nicely written. I find the topic interesting and timely. Thus, I recommend publication given the authors address the following minor comments, which are mostly regarding the figure style and data presentation:

Emax and IC50 appear in the abstract without definition.

A legend should be added to figure 1. In figure 1e, the data should be presented with an x-axis (log scale) to make the trend clear.

A legend should be added to figure 2

In figure 4a, the data should be presented with an x-axis (log scale) to make the trend clear.

In figure 6, the data should be presented with an x-axis (log scale) to make the trend clear.

In figure 7b, a “guide to the eye” would help to see the trend.

In general, the figures do not have the same style – some are boxed and some are not. The fonts and font sizes are different, and the sharpness is different.

Author Response

Dear Editor,

We would like to submit the revision of our manuscript, “Effects of Doxorubicin Delivery by Nitrogen-Doped Graphene Quantum Dots on Cancer Cell Growth: Experimental Study and Mathematical Modeling.” In this work, we used N-GQDs to effectively deliver doxorubicin to cancer cells and mathematical model to find time-independent IC50 values for the potential of improving chemotherapy dosing and effectiveness. This manuscript was reviewed by two referees who made minor comments in regards to figure style and data presentation, and required major revisions in regards to an appropriate analysis of the microstructure and chemical composition of N-GQDs. We are very grateful to the reviewers for the feedback and suggestions and introduced the following changes to the manuscript in order to address all of the comments in the point-by-point response:

Reviewer #1: This is a very interesting study by Frieler et. al. The authors assess the efficacy of Doxorubicin chemotherapeutic and compare the delivery method with and without nitrogen-doped graphene quantum dots (N-GQDs). The use of N-GQDs allowed using up to 10 times lower drug doses with the same therapeutic effect. The study is accompanied by a model of cancer cell growth. The work is well presented and the paper is nicely written. I find the topic interesting and timely. Thus, I recommend publication given the authors address the following minor comments, which are mostly regarding the figure style and data presentation:

Response to Reviewer 1 Comments

Point 1: Emax and IC50 appear in the abstract without definition.

Response 1: We appreciate the reviewer’s feedback. Emax and IC50 definitions have been added to the abstract in lines 12 – 14.

Point 2: A legend should be added to figure 1. In figure 1e, the data should be presented with an x-axis (log scale) to make the trend clear.

Response 2: We thank the reviewer for the comment. A legend has been added to figure 1e. The data presented in Figure 1e is now plotted on the log scale.

Point 3: A legend should be added to figure 2

Response 3: We thank the reviewer for the comment. A legend has been added to figure 2.

Point 4: In figure 4a, the data should be presented with an x-axis (log scale) to make the trend clear.

Response 4: We thank the reviewer for this comment as it made the graph more representative. The x-axis scale for both the concentration of doxorubicin and N-GQDs has been changed to the log scale.

Point 5: In figure 6, the data should be presented with an x-axis (log scale) to make the trend clear.

Response 5: We thank the reviewer for the comment. The x-axis is now changed to the log scale.

Point 6: In figure 7b, a “guide to the eye” would help to see the trend.

Response 6: We thank the reviewer for the suggestion. A guide to the eye has been made to the Figure 7b.

Point 7: In general, the figures do not have the same style – some are boxed and some are not. The fonts and font sizes are different, and the sharpness is different.

Response 7: We thank the reviewer for pointing out these deficiencies. The fonts of the figures have been adjusted and the major figures all plotted in Origin to be in the same style; additionally, all the graph figures have been boxed.

Reviewer 2 Report

In this work by Frieler and co-workers described how nitrogen-doped graphene quantum dots could be used to deliver doxorubicin to the cancer cells. They then made a comprehensive investigation of the impact of these drug-equipped nanoparticles on cancer cell growth. The results are impressive, interesting, and certainly fit the scope of the readership of Nanomaterials. The article's biological part has been conducted with appropriate scientific rigor, and the produced outcomes are valuable to the broader nanomaterials community. However, besides some minor shortcomings, the article lacks characterization of N-GNDs, which serve an active role in this solution. As a consequence, I suggest a major revision and invite the authors to supplement this study with an appropriate analysis of the microstructure and chemical composition of N-GNDs. Upon incorporating such crucial information, the article will reach a sufficient level to recommend its publication in Nanomaterials. The suggestions of the reviewer are given below:
1) State of the art is described very well, but the "in this research" section lacks the novelty statement. It is important to provide specific information on what is in this work that was not done before by others. This helps the readers decide whether to read the article but, most importantly, its impact can be easily evaluated. Please kindly include this in the revised version of the manuscript.
2) Headlines should not be separated from the corresponding paragraphs e.g. Line 164. Please fix it.
3) Graphene quantum dots are the heart of this article, but the article lacks their characterization. Providing microstructure and chemical composition is especially important in the case of nanomaterials, which show a very strong structure-property relation. Therefore, please include SEM/TEM images and possibly Raman spectra to judge the quality of nanocarbon.

Author Response

Dear Editor,

We would like to submit the revision of our manuscript, “Effects of Doxorubicin Delivery by Nitrogen-Doped Graphene Quantum Dots on Cancer Cell Growth: Experimental Study and Mathematical Modeling.” In this work, we used N-GQDs to effectively deliver doxorubicin to cancer cells and mathematical model to find time-independent IC50 values for the potential of improving chemotherapy dosing and effectiveness. This manuscript was reviewed by two referees who made minor comments in regards to figure style and data presentation, and required major revisions in regards to an appropriate analysis of the microstructure and chemical composition of N-GQDs. We are very grateful to the reviewers for the feedback and suggestions and introduced the following changes to the manuscript in order to address all of the comments in the point-by-point response:

Reviewer #2: In this work by Frieler and co-workers described how nitrogen-doped graphene quantum dots could be used to deliver doxorubicin to the cancer cells. They then made a comprehensive investigation of the impact of these drug-equipped nanoparticles on cancer cell growth. The results are impressive, interesting, and certainly fit the scope of the readership of Nanomaterials. The article's biological part has been conducted with appropriate scientific rigor, and the produced outcomes are valuable to the broader nanomaterials community. However, besides some minor shortcomings, the article lacks characterization of N-GQDs, which serve an active role in this solution. As a consequence, I suggest a major revision and invite the authors to supplement this study with an appropriate analysis of the microstructure and chemical composition of N-GQDs. Upon incorporating such crucial information, the article will reach a sufficient level to recommend its publication in Nanomaterials. The suggestions of the reviewer are given below:

Response to Reviewer 2 Comments

Point 1: State of the art is described very well, but the "in this research" section lacks the novelty statement. It is important to provide specific information on what is in this work that was not done before by others. This helps the readers decide whether to read the article but, most importantly, its impact can be easily evaluated. Please kindly include this in the revised version of the manuscript.

Response 1: We would like to thank the reviewer for pointing this out. The following has been added in lines 132 and 144:

So far, mathematical modeling approaches have been developed for the drug delivery of nanomaterials based on hydrogel [46], mesoporous silica [47, 48], and pegylated gelatin [49] that modeled for time-dependent drug concentration and tumor volume equations followed by the nanotherapy [50]. Such approaches are novel as they optimize the properties of the nanomaterial that have been often limited by clinical data and nanotherapeutics for clinical use [51], allowing to increase efficacy and/or repurpose the existing therapeutics [52]. The present study, however, presents a novel method of modeling drug delivery via a time-independent approach and for the first time provides a model for cancer therapeutic delivery by the graphene quantum dots. While experimentally GQDs have previously been conjugated with DOX for cancer targeted drug delivery [53, 54], the N-GQDs utilized in this work, possess higher biocompatibility, and allow for multicolor visible/near-infrared fluorescence imaging [27].

Point 2: Headlines should not be separated from the corresponding paragraphs e.g. Line 164. Please fix it.

Response 2: We thank the reviewer for the comment. Headlines have been corrected and placed above the corresponding paragraphs.

Point 3: Graphene quantum dots are the heart of this article, but the article lacks their characterization. Providing microstructure and chemical composition is especially important in the case of nanomaterials, which show a very strong structure-property relation. Therefore, please include SEM/TEM images and possibly Raman spectra to judge the quality of nanocarbon.

Response 3: We thank the reviewer for pointing out these deficiencies. TEM and EDX characterization as well as FTIR spectra of the GQDs have been provided in the supporting materials along with their description. Their discussion has been added to the text in the lines 279 – 285:

GQDs were characterized by the Transmission Electron Microscopy as 3-5 nm quasi-spherical structures with discernable graphitic lattice fringes (Figure S1). While EDX (Energy Dispersive X-Ray Analysis) shows the presence of carbon, oxygen and nitrogen within the GQDs (Figure S1), their FTIR (Fourier Transform Infrared Spectroscopy) spectra possess a number of features characteristic of the oxygen and nitrogen-containing functional groups decorating the surface of the GQDs. (Figure S2).

Round 2

Reviewer 2 Report

Thank you for following the suggestions. I recommend to publish the article in the present form.